# Localized Nanopore Fabrication via Controlled Breakdown

**DOI:** 10.3390/nano12142384

**Published:** 2022-07-12

**Authors:** Cuifeng Ying, Tianji Ma, Lei Xu, Mohsen Rahmani

**Affiliations:** 1Advanced Optics and Photonics Laboratory, Department of Engineering, School of Science &Technology, Nottingham Trent University, Nottingham NG1 4FQ, UK; lei.xu@ntu.ac.uk (L.X.); mohsen.rahmani@ntu.ac.uk (M.R.); 2Hunan Provincial Key Laboratory of Materials Protection for Electric Power and Transportation & Hunan Provincial Key Laboratory of Cytochemistry, School of Chemistry and Chemical Engineering, Changsha University of Science and Technology, Changsha 410114, China; tianji.ma@csust.edu.cn

**Keywords:** nanopore sensing, nanofabrication, controlled breakdown, single-molecule sensing, plasmonic nanopores

## Abstract

Nanopore sensors provide a unique platform to detect individual nucleic acids, proteins, and other biomolecules without the need for fluorescent labeling or chemical modifications. Solid-state nanopores offer the potential to integrate nanopore sensing with other technologies such as field-effect transistors (FETs), optics, plasmonics, and microfluidics, thereby attracting attention to the development of commercial instruments for diagnostics and healthcare applications. Stable nanopores with ideal dimensions are particularly critical for nanopore sensors to be integrated into other sensing devices and provide a high signal-to-noise ratio. Nanopore fabrication, although having benefited largely from the development of sophisticated nanofabrication techniques, remains a challenge in terms of cost, time consumption and accessibility. One of the latest developed methods—controlled breakdown (CBD)—has made the nanopore technique broadly accessible, boosting the use of nanopore sensing in both fundamental research and biomedical applications. Many works have been developed to improve the efficiency and robustness of pore formation by CBD. However, nanopores formed by traditional CBD are randomly positioned in the membrane. To expand nanopore sensing to a wider biomedical application, controlling the localization of nanopores formed by CBD is essential. This article reviews the recent strategies to control the location of nanopores formed by CBD. We discuss the fundamental mechanism and the efforts of different approaches to confine the region of nanopore formation.

## 1. Introduction

Nanopore sensing has been attracting attention rapidly over the past decades due to its appealing applications in label-free, real-time biosensing of single molecules in aqueous solutions [1,2,3,4,5,6]. A nanopore sensor detects the ionic current through an electrolyte-filled nanochannel connecting two reservoirs. Taking a synthetic nanopore as an example (Figure 1a), when an electrical potential is applied between these two reservoirs, the ions flow through the nanochannel and induce a measurable current that is directly dependent on the size of the nanochannel. A biomolecule diffusing through this nanochannel partially blocks the ionic flow, resulting in a detectable reduction in the ionic current. The red trace on the bottom of Figure 1a demonstrates an example signal of proteins translocating through a synthetic nanopore. Each blockade-current pulse corresponds to the translocation of the analyte through the nanopore. Figure 1b illustrates the current trace of a single protein transiting through the nanopore. The amplitude of the blockade current, Δ*I*, connects to the volume of the electrolyte solution displaced by the particle, thereby revealing the size of the target analyte. The duration of the current drop, or dwell time (*t_d_*), corresponds to the diffusion coefficient of the particle, providing information about the charge and size of the particle. With rigorous analysis, one can obtain information related to the particle shape, volume, charge, dipole moment, and diffusion coefficient [7,8,9,10,11,12,13,14,15,16,17,18,19,20]. Many reviews have discussed the applications and the full potential of nanopore techniques in DNA sequencing, protein fingerprinting, and virus detection [1,2,3,4,5,21,22,23,24,25,26,27,28,29].

There are two major categories of nanopores—biological nanopores and synthetic, solid-state nanopores. Biological nanopores consist of single transmembrane proteins inserted in lipid bilayers or polymer membranes. Commonly used biological pores, such as α-hemolysin (αHL) [30,31,32,33,34,35], *mycobacterium smegmatis* porin A (MspA) [36,37,38], *aeromonas hydrophila* Aerolysin (AeL) [39,40], and *bacteriophage phi29* (Phi29) [41,42], have relatively small constrictions with diameters smaller than 4 nm. The small constrictions of biological nanopores make them a perfect candidate for detecting small analytes such as metal ions, single polymer chains [35], DNA molecules [31,32,33,36,41,42], peptides [38,43,44], and unfolded proteins [30]. Moreover, these biological nanopores have a well-defined pore shape that is highly reproducible, and the proteins used to create biological nanopores can be harvested using synthetic biology techniques [2,3,6]. Since 2012, biological nanopores have been developed into portable and commercially available devices for DNA and RNA sequencing by Oxford Nanopore Technologies [45], and their potential has been further explored in the field of clinical and environmental research [46,47].

Despite these attractive advantages, biological nanopores suffer from several drawbacks [3,6]. Firstly, biological nanopores need to be inserted into suspended membranes formed by lipid bilayers or polymers. These membranes are usually mechanically fragile, allowing experiments to be performed only for a limited time. Moreover, biological nanopores are not suitable to be used in harsh environments such as extreme pHs, temperatures, or high electric fields. Lastly, the small diameter and fixed pore size of biological nanopores exclude their applications in analyzing large biomolecules such as proteins.

On the other hand, the diameter and geometry of synthetic nanopores can be tuned depending on the user’s requirement. These nanopores can be fabricated in a wide range of materials including silicon nitride [48,49,50,51,52,53,54,55,56], silicon dioxide [57], HfO_2_ [58], and polymers [59], as well as 2D materials including graphene [60,61], MoS_2_ [62,63], and BN [64]. The inherent advantages of solid-state nanopores make them suitable for integrating into nanodevices and offer flexibility in experimental conditions, even in extreme physical and chemical environments. In addition to the flexibility in material and size, a low-noise nanopore setup is crucial to detect the current drops (typically hundreds of picoamperes to nanoamperes) from each translocation event [65,66,67,68,69,70]. Such low noise is especially important at high bandwidth (>10 kHz), since most proteins and DNA bases translocate through a nanopore within several microseconds [6,11,67,71,72]. An ideal nanopore should be fabricated in a low-noise, low-capacitance substrate with adjustable size to match the analyte in order to optimize the signal-to-noise ratio.

Recent developments in nanofabrication allow a wide variety of approaches to fabricate such pores, including but not limited to transmission electron microscopy (TEM) drilling [48,49,50], focused Helium/Gallium ion beam drilling [51,52,53,54], fused silica capillary shrinking [73,74], gold nanoparticle heating [75,76], ion-beam sculpting [55,56], reactive-ion etching [77,78], and ion track etching [59]. Several reviews have discussed the fabrication of nanopores in depth [27,79,80]. Despite the diverse approaches in fabricating nanopores, obtaining ideal synthetic nanopores remains one of the most time consuming and expensive processes in the field of nanopore sensing. All the abovementioned approaches require special expertise, cleanroom facilities, and expensive equipment to fabricate nanopores and most of them cannot control the pore size precisely.

Due to the barriers to fabricating nanopores mentioned above, the controlled breakdown (CBD) method of nanopore fabrication immediately attracted wide attention since it was reported in 2014 by the Tabard-Cossa group [81]. CBD uses the configuration of the nanopore setup to apply a high voltage/electric field across a thin insulating membrane that, at first, contains no pores. The electrical stress introduces defects randomly in the insulating membrane and eventually, the defects connect to form a physical path at the nanometer scale [82]. By monitoring the current across the membrane as these defects accumulate, CBD can detect this pore-formation event. This breakdown principle allows CBD to generate nanopores precisely using a DC voltage generator and current feedback, which significantly reduces the price of nanopore fabrication. Moreover, the nanopores are fabricated in situ, so the resulting nanopores are immediately ready to use for biosensing. In 2021, a commercially available nanopore fabrication system, named Spark-E2, was created to generate nanopores automatically with sub-nm precision of the target size. This device, from Northern Nanopore Instruments, can generate pores with diameters of 1 nm to 20 nm within 1 h, with a success rate larger than 85% [83].

Despite of all these advantages, CBD nanopore fabrication comes with some disadvantages in comparison to other fabrication techniques. Firstly, multiple nanopores can be formed by CBD. During the past five years, several works have attempted to reduce the formation of multiple nanopores by different approaches [10,83,84,85,86,87,88,89,90,91,92]. The recent review by Fried et al. has summarized the progress in this area thoroughly [93]. In addition to the randomness in the nanopore numbers, CBD offers no control over the nanopore location in the membrane [10,87,88,90,94,95]. The random location of nanopores in the membrane, however, does not matter for traditional nanopore sensing due to its sensing principle—resistive pulse sensing.

In recent years, combining nanopore sensing with fluorescence and plasmonic optical measurements has drawn more and more interest, because optical signals provide additional depth of information without interfering with the electrical signal [22,96,97,98,99,100,101,102,103,104,105,106,107,108]. In addition, the research field of plasmonic optical sensing, or fluorescence-based single-molecule sensing, could benefit from the electrical signal of nanopore sensing. Nanopores can assist the capture rate of biomolecules for nanophotonic sensing, while at the same time, the electrical signal from nanopore sensing can reinforce the single-molecule signal from optical measurements [95,100,109,110,111,112]. CBD, however, faces challenges in forming nanopores in a precise position for optical sensing. In this review, we discuss the state-of-the-art of how CBD is used to fabricate nanopores for various sensing applications. The review focuses on recent strategies to improve the local confinement of nanopores formed by controlled breakdown. We review these approaches with an insight into their advantages, challenges, and applications.

## 2. CBD Nanopore Fabrication

### 2.1. Principle of Forming Nanopores by Breakdown

When a dielectric material is under high electric field stress for a certain duration, the material loses its insulating properties locally. This process, or the failure of dielectric materials under electric field stress, is known as dielectric breakdown [113,114]. Most research has focused on understanding the mechanism and the dynamics of breakdown to improve the reliability and the performance of electronic devices. Two types of breakdowns can occur in thin membranes: (i) high external voltage breakdowns and (ii) low external voltage breakdowns. The high voltage breakdowns are an intrinsic process related to the defects building up to a critical level under external stress, which is a stochastic process. On the other hand, low external voltage/electric field breakdown is generally attributed to extrinsic defects, such as extended defects in the substrate that produce localized property changes in the material. These localized changes could be due to dielectric thinning, temperature differences, oxide defects, or other properties that locally increase conductivity.

Similar to the breakdown of thin films in a dry atmosphere, breakdown can occur in a liquid environment when a thin membrane is under external electric field stress. Instead of aiming to avoid breakdown, Tabard-Cossa’s group took advantage of this stochastic process for nanopore fabrication, which is now widely known as controlled breakdown (CBD) [81]. Figure 2a illustrates the steps of the CBD process in a SiN_x_ membrane. An external electric field across the membrane, typically 0.6–1 V/nm, induces defects inside the membrane due to electric field stress. These defects increase the leakage current tunnelling through the dielectric membrane and accelerate the accumulation of defects (also known as charge traps). When the density of defects is high enough to form a connecting path in the insulating membrane, the current tunnelling through this path locally removes the membrane material in a short time, creating an open channel inside the membrane. Nanopores with an initial size distribution from 0.5 to 3 nm can be formed if the external electric field is removed immediately after the sudden increase of current. This process is the fabrication phase of CBD, and only one nanopore is expected to form during this phase.

For protein characterization, there is a need for a vast variety of nanopore sizes, ranging from several nanometers to one hundred nanometers. After the initial breakdown, further applying an electric field with feedback control allows for the expansion of and precise control over the nanopore size. During this conditioning phase of CBD (Figure 2a), the applied electric field strength is typically one-third of the electric field strength during the breakdown, and it should never exceed this [83]. The channel shape can also be adjusted slightly during this phase [115]. It is expected that the nanopore will expand to the desired size during the conditioning process. However, the enlargement process is fundamentally different from the breakdown one [92]. Multiple nanopore formation has been reported during this conditioning phase. In addition, soft breakdown without thorough pore generation can also occur, leading to so-called “fake” nanopores [90,116,117].

CBD has been rapidly applied in fabricating nanopores within different materials and in different sizes. Figure 2b shows that CBD can fabricate nanopores in SiN_x_ membranes with diameters ranging from single nanometers to several tens of nanometers [10,91,118]. By forming nanopores slightly larger than the target analytes, CBD could efficiently optimize the signal-to-noise ratio of nanopore sensing. In addition to size control, CBD also makes it possible to generate nanopores in several materials that are otherwise challenging to use with traditional nanopore fabrication techniques such as energetic nanoparticle drilling. For instance, it is difficult to drill nanopores in a SiN_x_ membrane on a fused-silica substrate, because the charged particles used for drilling cannot be conducted away. CBD, on the other hand, can easily form nanopores in this substrate, which enables the production of low-noise nanopore devices [72]. CBD has proved its ability to form nanopores in membranes composed of various materials such as SiO_2_ [57], HfO_2_ [58], and metal-insulating layers [120,121], as well as in 2D materials including graphene, BN, and MoS_2_ [119,122,123,124,125,126,127,128,129].

### 2.2. Approaches to Controlled Breakdown

Despite its many applications, fabricating nanopores by CBD comes with drawbacks due to the intrinsically random nature of the breakdown process. The issues of the CBD technique that need to be addressed include: (1) the possibility of a long and uncertain amount of time necessary to fabricate a nanopore; (2) false breakdown events due to soft breakdown; (3) the potential to form multiple nanopores in the membrane that are difficult to identify using electric current feedback; (4) pore formation in random locations in the membrane; and (5) the impossibility of using CBD to form nanopore arrays.

An ideal controlled breakdown protocol would meet the following criteria to allow researchers to benefit from its low cost and broad accessibility: (1) a short time-to-breakdown to allow efficient pore fabrication; (2) precise size control of the generated nanopore, including both a tight and small size distribution of initial nanopores in the fabrication phase, as well as an efficient feedback control over the pore size during the conditioning phase; (3) the production of single nanopores in the membrane rather than forming multiple nanopores; (4) a smooth surface and regular nanopore shape to allow a high signal-to-noise ratio for nanopore sensing; and (5) control over the localization of nanopores formed in the membrane.

In the context of dry membranes, the time-to-dielectric breakdown is strongly dependent on the parameters of the material used, including the thickness of the membrane, the composition of the material, and the thin-film deposition process. In addition to the intrinsic nature of the materials, several external conditions also contribute to the time-to-breakdown of the thin membrane, including the electric field strength, external induced defects, and temperature. Tabard-Cossa’s group analyzed the stochastic breakdown process of SiN_x_ membranes with different thicknesses by applying a high constant electric field while measuring the time-to-breakdown [81]. The distribution of breakdown time indicates that the probability of forming a nanopore with randomly generated defects throughout the membrane can be described using the Weibull distribution [116]. Thus, two identical membranes can break down at different times when subjected to the same electric field. To control the efficiency, quality, and quantity of the nanopores formed by CBD, all parameters that influence the breakdown and enlargement process should be considered.

Here, we discuss the parameters influencing the breakdown in both dry and liquid conditions, as well as the parameters that only apply to the liquid environment, such as the pH and salt concentration of the electrolyte.

#### 2.2.1. Material Composition of Membrane

Research on thin membranes of electronic devices shows that different materials exhibit different breakdown properties [113,114,130,131,132,133,134], the stoichiometry of the SiN_x_ thin film affects the breakdown process strongly due to the changes in electrical conductivity associated with the Si content [86,130,131,132]. Such material-dependent breakdown has also been observed in nanopore fabrication via CBD [81,85,91,120]. Si-rich SiN_x_ is often used for nanopore sensing due to its excellent mechanical stability, allowing it to be fabricated into large-area free-standing membranes [135,136]. The use of Si-rich SiN_x_ membranes is favorable for CBD fabrication, because high silicon to nitride ratios lead to a high density of charge traps [81,85,91,120]. As a trade-off, nanopores in Si-rich SiN_x_ membranes tend to expand and grow over time when immersed in a salt solution [137]. Recently, Fried et al. [121] studied the breakdown process of different compositions of SiN_x_ membranes when voltage is applied via electrolyte solution, electrodes, and the combination of both. For Si-rich SiN_x_ membranes, a high voltage is required to induce breakdown, because the electrochemical reaction between the electrolyte and dielectric interface limits the conduction across the membrane.

In addition to the SiN_x_, CBD has been used for fabricating nanopores in other insulating materials such as SiO_2_ [57] and HfO_2_ [58]. Yun et al. compared the kinetics of breakdown for SiO_2_, SiN_x_, and HfO_2_ by investigating their breakdown potential using linear sweep voltammetry [138]. They found that the electrochemical nature of different materials influences their breakdown kinetics. The breakdown potential of SiN_x_ is not sensitive to cations such as sodium or silver ions. These ions, however, have a significant influence on the breakdown potential of SiO_2_ membranes. HfO_2_, on the other hand, is resistant to sodium ions but strongly affected by silver ions, likely due to the intrinsic oxygen vacancies. In addition to pure insulating materials, CBD has demonstrated its ability to generate nanopores in metal-insulating layers [120,121]. This capability provides a new path for integrating nanopores into nanodevices with a solid-state nanopore perfectly aligned with the on-chip electrodes [84], which is otherwise challenging for traditional approaches of nanopore fabrication.

#### 2.2.2. Membrane Thickness

Most nanopores fabricated by CBD are in SiN_x_ membranes with a thickness between 10 to 30 nm with a breakdown field strength typically between 0.6 to 1 V/nm [55,64,66,67,70,75,76,77,111,132,139]. The breakdown process can be characterized overall by monitoring the leakage current due to the charge transport through defects in the dielectric material. This tunnelling current accelerates the defect accumulation and ultimately leads to a breakdown, which is indicated by a sudden increase in current. The charge transport can only tunnel through thin membranes, so a thick membrane might not be suitable for nanopore formation via CBD. The thickest membrane reported for CBD without any additional assisting technique is 50 nm [94]. Beyond that thickness, it is difficult to form a tunnelling current for a breakdown unless defects are introduced locally via external assistance. The leakage current across thinner membranes, on the other hand, is easy to form when applying an external electric field. Yanagi et al., reported the gradual increase in leakage current across thin membranes with a thickness thinner than 5 nm [140], which is similar to “soft breakdown”. A slow increase in the current during CBD allows the formation of small nanopores with diameter of 1 to 2 nm [118,140].

#### 2.2.3. Electrolytes

In addition to the above parameters, liquid-contacting membranes have some unique parameters that also affect the breakdown process. When the solid material is in contact with liquid, the electrostatic field on the surface of the solid material attracts the counterions in the liquid, forming an electrical double layer at the solid–liquid interface. When there is a potential difference across the interface, electrochemical reactions occur at the interface, which induces the injection and/or removal of charges in the material [121]. The interface between the electrolyte and the insulating membrane is the cause of the complexity of CBD. Matusi et al., reported that a membrane that initially did not have defects exhibited defect formation after an electrolyte was brought into contact with the membrane [82]. Several research works have demonstrated the correlation between the breakdown kinetics of the membrane and the pH of the solution [85,91,116]. Breakdown happens faster in extreme pH conditions, either strongly acidic or strongly alkaline, indicating the strong involvement of the hydrogen and hydroxide ions in the breakdown kinetics.

Other parameters, such as solute concentration and solvent type, have also been shown to affect the breakdown process [81,87,89,116,138,141]. A recent report has pointed out that, during the conditioning phase, nanopore enlargement exhibits weaker dependence on the pH of the electrolyte, because the high ionic current in the formed pore accelerates the electrochemical reactions on the nanopore wall [92]. Using a high salt concentration in the conditioning process can increase the ionic current through the nanopore and, therefore, could expand the nanopore efficiently while not risking the generation of another breakdown path [83,92].

#### 2.2.4. External Electric Field and Feedback Controls

There are two feedback controls required in CBD nanopore fabrication. One is the detection of the breakdown, and the other is the termination of the external electric field during nanopore enlargement. Depending on the applications, different approaches are used and/or combined with feedback control to optimize nanopore fabrication by CBD.

**Detection of a breakdown event.** An efficient feedback control system allows a high yield rate of nanopore fabrication and a tight size distribution of initial nanopores. The simplest feedback control is to use a cut-off threshold when applying constant electric field stress [81,142]. This threshold feedback allows one to stop the applied electric field as soon as the measured current (or voltage in the case of applying constant current) exceeds the pre-set value. This approach requires minimal programming and simple instrumentation. During this process, the time-to-breakdown is the only parameter changed and, therefore, is often used to test the mechanism of breakdown across different conditions such as different pHs and salt concentrations of the solution. The drawback of this threshold cut-off is that the distribution of the resulting pore sizes is wide if the feedback time is not fast enough or if the external electric field is high. Moreover, a soft breakdown could be picked up as a breakdown event. An alternative feedback control method is to use a sudden increase of current as a criterion for nanopore generation. Such feedback control can avoid the problem of overgrowing the nanopore, but soft breakdown could still be falsely detected as a pore generation [85,90].

To differentiate a soft breakdown from a hard breakdown, Roshan et al. [117] developed an algorithm called the moving-z-score-based CBD, to detect the anomalous points in the current–time trace while applying the voltage. The reported value, or z-score, considers the standard deviation and the mean value of the measured current within a certain window size. Compared to the threshold cut-off, this method minimizes the detection of false breakdowns and hence increases the yield rate. It also generates a tight distribution of the pore sizes since it detects a sudden change of current rather than a cut-off value.

Instead of applying a constant electric field across the membrane, other protocols use ramping voltage pulses or constant currents to create an external electric field stress. The strategy of ramping the voltage can be very useful and efficient for membranes from different batches, e.g., batches with different thicknesses or material compositions [10,83,140]. This process is often combined with leakage feedback to stop the increase of the voltage. Note that there is no conclusive correlation between the shape of the applied electric field with the initial pore size, although this parameter can have a large impact on the fabrication time.

**Termination of the breakdown process when the pore meets the desired size.** After the detection of the initial breakdown, a CBD protocol usually reduces the applied electric field to slowly expand the nanopore to the desired size. To estimate the pore size during the enlargement process, one can measure the resistance across the membrane by measuring one of the current, the voltage, or the IV curve during the breakdown. The accuracy of the pore size and the time of the feedback are two key features to ensure the size control of nanopores.

This conditioning phase carries the risk of potentially generating multiple nanopores in a membrane [10,58,90,94,143]; hence, an efficient protocol at this conditioning phase is essential to achieve relatively large nanopores (i.e., >10 nm). Many groups have demonstrated that low electric fields have a higher success rate than high electric fields in forming single nanopores in a membrane [10,81,83,85,94,116]. The electrical feedback methods, however, assume that only one nanopore is formed in the membrane. One of the major concerns in CBD nanopore fabrication, however, is the formation of multiple nanopores in the membrane. For this reason, being able to detect the quantity of nanopores in a single membrane in situ is highly useful.

#### 2.2.5. Additional Non-Electrical Feedback Controls

Several approaches have been proposed to monitor the nanopore enlargement process in addition to the electrical signal feedback. Measuring the conductivity change after coating the membrane with a self-assembled layer can report whether a single pore is formed in the membrane [10,122]. This approach not only reveals the pore number, but also indicates the shape of the formed nanopores [10,144]. However, although this approach can be used for in-situ examination for pores right after the breakdown, the “resolution” of the discrimination is quite coarse due to the possibility of the partial coating of the nanopore wall or complete occlusion of the nanopore entrance by the coating layer.

On the other hand, optical signals can report the exact number of pores formed in a membrane. Optical measurements also have no interference with the electrical signal and hence can be implanted in a system as an orthogonal source of information. Three different optical signals have been explored for monitoring nanopore formation during the breakdown in real time: (1) photoluminescence (PL) intensity [86,88,89,145], (2) measuring ionic current while laser scanning [87,95], and (3) Ca^2+^ ions and Ca^2+^ indicator [10,88,90]. We will discuss these optical feedback controls in Section 3.2.

All optical measurements as feedback controls come inherently with low spatial resolution due to the optical diffraction limit. This resolution cannot discriminate multiple nanopores that are formed in the focus area, e.g., Figure 4 of Ref. [85]. In addition, an optical signal cannot reveal the shape nor the channel length of nanopores. Therefore, TEM and AFM remain important tools to characterize nanopores at nanometer resolution.

## 3. Localized Nanopore Fabrication by CBD

Optical nanopores [96] can benefit from the high bandwidth of optical measurements, as well as the ease of monitoring optical signals at high throughput [22,96,97,98,99,100,101,102,103,104,105,106,107,108,146,147,148]. On the other hand, nanopore techniques can actively deliver target molecules into the optical sensing area instead of relying on diffusion or tethering [95,105,110,112,149]. Given the advantages of CBD nanopore fabrication, it could be very attractive if CBD could generate nanopores at specific positions in a membrane. Since the CBD technique is relatively new compared to nanopore sensing, attempts to generate localized nanopores for optical measurements are still in progress. As discussed in Section 2.1, the extended defects in the substrate produce localized property changes in the material, promoting the probability of nanopore formation in this region. Such localized changes of material could be introduced by dielectric thinning, temperature differences, oxide defects, or other properties that locally increase in conductivity. This section will discuss the approaches that assist the formation of nanopores in a specific region, including laser-assisted breakdown [86,88,89,144], pre-thinning [150], local high electric field [151,152], and local liquid contact [153,154,155,156].

### 3.1. Laser-Induced Breakdown for Localized Nanopore Fabrication

One year after CBD was reported, the Dekker group first reported a self-aligned nanopore formation at an optical hotspot using laser-induced breakdown [95]. They demonstrated that the laser-assisted breakdown formed a nanopore locally at the area where the optical field was enhanced, which they created using a plasmonic bowtie structure. Since then, laser-assisted breakdown has attracted dramatic attention. In 2018, three different groups reported the localization of pore formation by using laser etching or laser-assisted breakdown [10,88,89]. Further research focused on the fundamentals of laser breakdown or applying laser-assisted breakdown to fabricate nanopore arrays [86,87].

Fundamentally, the laser-induced breakdown process involves three different effects accelerating pore formation: (1) laser-accelerated defect accumulation, (2) laser thinning, and (3) thermal-assisted etching/breakdown. Although the fundamentals of laser-assisted breakdown can vary between experiments, the setup of laser-assisted breakdown remains similar. Figure 3a illustrates schematically a typical laser-assisted breakdown setup. In general, a laser beam is focused on the SiN_x_ membrane with a spot size of a micrometer or less. An additional optical feedback control (discussed in Section 3.2) either photoluminescence from the SiN_x_ membrane, or Ca^2+^ flux fluorescence microscopy can be added to monitor the pore-formation process.

Whether we detect a thinned membrane or defect accumulation of the membrane depends on which fundamental mechanism governs the pore-formation process. The governing mechanism for accelerated pore formation with the assistance of laser illumination depends on the laser wavelength, the material of the membrane and its surrounding environment. Through different characteristic approaches, one can figure out which mechanism governs pore formation.

**Laser-accelerated defect accumulation.** When a laser illuminates the dielectric membrane, it promotes the rate of defect accumulation and hence induces a higher leakage current [10]. Such laser-accelerated defect accumulation promotes pore formation in the region of the membrane that is illuminated by the laser. As illustrated in Figure 3b(i), laser irradiation increases the free-electron density in the conducting band of dielectric materials, thereby increasing the density of the defect locally in the laser spot [157]. These defects are visible in the TEM image as a dark spot in the membrane where the laser spot was focused, as shown in Figure 3b(ii) [10]. Both the reflective microscopy image (Figure 3b(iii)) and the transmission microscopy image (Figure 3b(iv)) exhibit a dark spot on the membrane where the laser was focused, indicating a change in the material’s optical properties after laser illumination rather than a thickness change. Since the defects inside the membrane facilitate the leakage current and hence the breakdown process, the time-to-breakdown will be dramatically reduced.

**Laser thinning.** The laser etching of a SiN_x_ membrane is a well-established technique for nanopore fabrication. This approach relies on the photochemical destabilization of Si–Si bonds in a Si-rich SiN_x_ membrane. Combining CBD setup with laser thinning provides in-situ nanopore formation, which can be beneficial for integrating the nanopore with microfluidics and biosensing (this aspect is well summarized in a recent review by Fried et al. [93]). During the laser thinning process, Si forms transition compounds with the anions in the solution and quickly becomes oxidized to SiO_2_ that hydrolyzes to silicic acid and dissolves in the electrolyte solution. Therefore, laser-thinning CBD has a strong dependence on the Si:N composition and is accelerated in alkaline solutions [86,87,158]. This photo-active etching results in a locally thinned membrane, where the electric field strength is much higher than that in other regions, promoting the breakdown in this laser-thinned region [86,88,89,90,145,159]. Figure 3c(i) illustrates the CBD process in combination with laser thinning. The thickness of the membrane during the laser-thinning process can be monitored by the photoluminescence (PL) of the SiN_x_ detected by an avalanche photodiode (APD), as shown in Figure 4a. In addition to the PL intensity, the thinned region appears bright in TEM images (Figure 3c(ii)), opposite to the observation for laser-accelerated defect accumulation. The resulting reflective microscopy image in Figure 3c(iii) shows a dark spot in the laser-thinned region, while the transmission microscopy image shows a bright spot due to the thinner area. The AFM profile of the membrane surface also confirms the existence of this thinned region on the SiN_x_ membrane [87,89].

**Thermal-assisted etching/breakdown**. Both laser-induced defect accumulation and laser-thinning processes are accompanied by a temperature increase due to laser heating. When the nanopore is illuminated by a laser beam, the membrane absorbs the laser radiation and converts it to joule heating. This localized laser-induced heating accelerates the reaction rate of either the breakdown process or the thinning process. In addition to its role in thermal-assisted etching/breakdown, laser heating has been used for nanopore fabrication, e.g., the photon-to-heat graphene nanopore sculpting [160] and the thermal shrinking/annealing of nanopores [161,162].

Electron and light microscopy images indicate an irreversible change in the membrane material after laser illumination, either due to laser-induced defect cumulation or laser thinning. The dominant mechanism varies among experimental conditions, e.g., the laser wavelength, applied voltage, and electrolyte solution. Generally, laser thinning occurs when the membrane is illuminated by a blue or green laser (488 or 532 nm), while a near-infrared laser normally introduces defect cumulation inside the membrane. The dominant mechanism for pore formation can be characterized by transmission light microscopy—a laser-thinning region appears bright while laser-induced defect accumulation appears dark, as shown in Figure 3b,c(iv). TEM and AFM images can also provide information regarding the fundamental mechanism of pore formation. These methods, however, remove the in-situ nature of laser-assisted CBD. Tang et al. proposed a statistical model to discuss the efficiency of using a laser to confine the location of a breakdown nanopore [87]. Their model predicted that the combination of high laser power and a low electric field generally exhibits the highest confidence of forming a nanopore at the laser-focused spot.

In addition to the localized pore formation, laser-assisted breakdown also allows the fabrication of nanopore arrays [86,87,88,89,90,159] and self-aligned nanopore formation [95]. Laser-assisted breakdown for nanopore array fabrication, in particular, will be of use for plasmonic nanopore applications [95,99,100,109,110,111,163,164,165,166,167,168,169,170]. For example, the use of a zero-mode waveguide is often hindered by a low capture rate [171]. A nanopore could help deliver the target DNA to the sensing zone and increase the fluorescent detection efficiency. Another example of how optical measurement can benefit from the nanopore technique is the plasmonic optical tweezer, which has a low trapping rate on proteins but shows improved trapping efficiency when a nanopore is introduced to the hotspot [100,110,172,173].

### 3.2. Optical Readout to Characterize the Localization of CBD Nanopores

The integration of an optical setup with CBD provides a unique platform to characterize nanopores formed by CBD in real time. As discussed in Section 2.2.5, there are three major approaches to monitoring the laser-assisted CBD process in situ, as illustrated in Figure 4.

**Photoluminescence (PL) intensity.** The photoluminescence (PL) is the light emission from materials [174]. The PL from a SiN_x_ membrane covers a spectral range from UV to IR [175,176,177]. The emitted PL intensity corresponds to the local thickness of the membrane. During a laser-thinning experiment, the photon count of the PL intensity collected either by a spectrometer or by an EMCCD offers real-time feedback of the membrane thickness at the laser-focused spot [88,89]. When a nanopore is formed (i.e., the thickness of the membrane is zero), the photon count of the PL drops dramatically. As shown in Figure 4a, a drop in PL photon counts and an increase of the current through the membrane indicate the formation of a nanopore. Although the PL intensity provides additional feedback on the pore-formation process, it does not indicate any information on the pore size, since the laser spots are typically of the order of several hundred nanometers. By scanning the PL over the membrane, one can obtain the location as well as the number of nanopores formed by laser etching [86,88,89,145,159].

**Measuring ionic current by laser scanning.** This approach relies on the increased ionic conductivity of the electrolyte due to local heating when a laser is focused on the nanopore location. The membrane, often silicon nitride, absorbs the partial laser energy and converts it to heat. This heat cannot disperse well due to the boundary between the membrane and the liquid, resulting in local temperature enhancement. Figure 4b(i) depicts the temperature profile when the laser is focused on a membrane with a nanopore. Laser-induced heating can establish a well-controlled temperature inside the nanopore [145,162,163,166,178,179,180,181,182,183,184,185,186,187,188]. This temperature control can be used for increasing the capture rate of the nanopore [166,183,184,187], studying the behavior of biomolecules at controlled temperature levels [163,167,178,180,181,183,184,186,189], and characterizing laser profiles with nanoscale resolution [168,185]. The laser heating of the SiN_x_ membrane increases the temperature of the electrolyte in the nanopore. As the temperature increases, the viscosity of the electrolyte decreases [10,87,89,95,190,191], resulting in high conductivity across the membrane. Figure 4b(ii) shows that the ionic conductivity of a nanopore increases upon laser illumination and drops to a baseline level in the absence of laser illumination [10]. Therefore, an increase in the ionic current during laser scanning indicates the overlapping of the laser spot with the nanopore [87,95]. Since the temperature of the electrolyte in the nanopore only increases when the nanopore is within the focus point, this method can only detect a nanopore with a spatial resolution of one µm [86,87,88,95,145].

**Ca^2+^ flux detection.** Monitoring the Ca^2+^ concentration near the nanopore by using a Ca^2+^ indicator provides an optical signal to detect ion flux through a nanopore [99,192,193,194]. This approach detects the fluorescence intensity of a Ca^2+^ indicator dye (Fluo-8 or Fluo-4) changing in response to the concentration of Ca^2+^ ions [195]. The fluorescence efficiency of the Ca^2+^-indicator increases when it binds to the freely diffusing Ca^2+^ ions, providing information about the local concentration of Ca^2+^ ions near the nanopore. Figure 4c illustrates the concept of the Ca^2+^ nanopore detection method. Ca^2+^ ions and Ca^2+^ indicator dyes are separated by the membrane with the nanopore as the only connection. When the applied voltage is negative, the electrophoretic force pulls Ca^2+^ ions by diffusion from the *trans* to *cis* chambers through the nanopore. When the polarity is reversed, the Ca^2+^ is driven from *cis* to *trans* through the nanopore and binds to the Ca^2+^ indicator dyes on the other side of the reservoir. The Ca^2+^ indicator then emits fluorescence that is localized near the entrance of the nanopore. Monitoring the Ca^2+^ ion flux through nanopores enables an optical readout to provide real-time, widefield feedback of the quantity of nanopores formed in the membrane [10,90]. This approach has been reported for revealing the location of nanopores formed during CBD, as well as multiple nanopore formation.

### 3.3. Tip-Induced Breakdown (AFM, Pipette)

In addition to the above methods, tip-induced breakdown offers another way to generate nanopores locally in a membrane and reduces the possibility of forming multiple nanopores. This group of methods relies on pre-defining a small region with either HIM, an AFM tip, or a micropipette. This small region limits the area of pore formation to a much smaller space than the whole free-standing membrane.

Using a pre-defined area to form nanopores locally by CBD was first reported by Carlsen et al. [150]. They used HIM to create a thinned area as small as 100 × 100 nm^2^ on a low-stress SiN_x_ membrane with a window size of 100 × 100 µm^2^ (Figure 5a). A 90% thickness reduction makes the electric field 10 times higher in the thinned region than the electric field strength across the rest of the membrane. This method can not only control the pore positioning, but also largely reduces the pore formation times for a given applied voltage. Such CBD at a pre-thinned membrane would be especially significant for a membrane that was originally quite thick. Though this method offers high-precision pore positioning for CBD fabrication, the use of helium ion beam milling counteracts two of the main advantages of CBD, which is its low cost and ease of accessibility.

Zhang et al. [153] further pushed the precision of the pore position to tens of nanometers using an AFM tip-controlled local breakdown (TCLB) (Figure 5b). An AFM tip (around 10 nm in diameter) is first set over the membrane. Then, the tip approaches the membrane at a speed of ≈5 μm s^−1^, and finally engages the membrane surface with a small loading force of 1 nN. After applying a voltage pulse with an amplitude of 24 V and a pulse duration of 100 ms, a nanopore is formed at the tip location. A log-normal probability distribution of the time of pore formation reveals that TCLB allows a lower degree of randomness for pore formation compared to the classic CBD. More interestingly, AFM provides an immediate characterization of the newly formed pores in situ (in the case of forming a pore diameter larger than the tip diameter). This method offers the most precise position control of any currently known CBD technique and largely reduces the possibility of forming multiple nanopores. The use of AFM, however, increases the cost of pore fabrication, and the alignment processes can be time consuming.

Similarly, liquid meniscus contact using a glass pipette can also form nanopores in a precise position [154]. As illustrated in Figure 5c, a glass micropipette filled with electrolytes is held by a micromanipulator above the membrane in an air environment. The pipette then moves down until liquid contact with the membrane is formed. This liquid contact generates a contact area of 1 µm in diameter, which can be confirmed by capacitance feedback. A nanopore is formed within the contact area by applying a breakdown voltage between the electrolyte in the pipette and the electrolyte beneath the membrane. The micromanipulator makes it possible to repeat the CBD process many times per membrane, therefore enabling the fast preparation of nanopore arrays. Based on this principle of pipette tip positioning, Yin et al. developed a transient high-electric-field controlled breakdown (THCBD) [151] by applying a high voltage before establishing the liquid contact (Figure 5d). Their work demonstrates that pore-forming time is inversely dependent on the applied voltage, while pore diameters have a linear relationship with breakdown voltage. Benefiting from the high voltage (several V/nm), they achieved pore-forming times in the range of several milliseconds, offering a new strategy for the fast fabrication of nanopore arrays. Despite these advantages, the contact area of 1 µm in diameter is relatively large compared to the nanopore size, which limits the pore positioning ability in this method.

In addition to using tip-based confinement, the electric field can also be confined by electrolytes within microfluidic systems to fabricate nanopores precisely, as illustrated in Figure 5e. Such a microfluidic system can select the fluidic channel to apply a voltage across different areas, allowing nanopores to form in selective channels [155,156]. By using this approach, Tahvildari et al. demonstrated the fabrication of five nanopores inside the microfluidic system. This liquid contact can also be introduced by a droplet using a pipette tip. Recently, Wang’s group demonstrated that the applied voltage can be controlled across different membranes and generate nanopores at a specific position in the flow system [155,156,197]. They also reported that CBD nanopores formed in a localized manner at the cross-disjoint mortise structures generated by Ga-FIB etching [198].

Linaro’s group combined chemical etching with CBD to fabricate high aspect ratio silicon nanopore arrays [196]. They deposited a silicon wafer with a thickness of 2 µm on a thick support (around 500 µm) and then mounted it between two chambers, filled with 5% HF: H_2_O: C_2_H_5_OH (1:8:3) solution (cis) and NaCl solution (9 mg/mL) (trans), respectively. The high external electric field accelerates the generation of holes (h^+^) in silicon, which collides with the crystal lattice to free the bound electrons. When the current density is high enough, the following electrochemical reactions occur:Si + 2*F*^−^ + 2h^+^ → SiF_2_
SiF_2_ + 2HF → SiF_4_ + *H*_2_

As illustrated in Figure 5f, a pyramid shape is formed by the chemical etching. Then the voltage is decreased and maintained at a constant low level. The constant voltage maintains the breakdown etching selectively at the sharpest tip to ensure a minimized pore size. The pore diameter can be controlled by the applied bias profile to satisfy different applications. This method opens a new way to fabricate single nanopores and nanopore arrays with a low chance of having multiple nanopores. The high aspect ratio of the formed nanopores, however, limits their sensitivity in biomolecule sensing.

Very recently, Fried et al. demonstrated the pore formation locally at the area of metal electrodes by locally injecting electrons into the membrane. They deposited metal electrodes on the surface of the SiN_x_ membrane [121]. These metal electrodes provide the electrons locally to assist nanopore formation in the area covered by the electrodes. Their work provides a fundamental understanding of the mechanism of nanopore formation during CBD, as discussed in Section 2.2. Fried et al. [84] have also reported a new CBD strategy to fabricate multiple nanopores locally using on-chip electrodes. The nanopores are self-aligned at the position where an on-chip electrode and electrolyte solution are in contact with the opposite side of the membrane.

## 4. Conclusions and Outlook

Since the nanopore-sensing concept was proposed 20 years ago, the nanopore technique is now at the stage of moving toward real-world applications and marketing. One main challenge to this progress, however, remains the fabrication of ideal nanopores with precisely controlled sizes, shapes, and locations. Controlled breakdown (CBD) has greatly improved the nanopore-sensing field by providing a low-cost and broadly accessible approach. Localized nanopore fabrication via CBD will further push the nanopore technique to the broader biosensing community. Future development of CBD techniques might focus on addressing questions/obstacles of CBD techniques such as the breakdown mechanism of different membrane materials, improving the long-term stability of CBD nanopores, and minimizing the surface roughness of the pore channel. All the above aspects are crucial for CBD techniques to be integrated into different nanodevices. Combining CBD with advanced techniques, such as nanofluidics, nanophotonics, AFM, or field transistors with 2D materials, will allow nanopore sensing to move closer to being a ready-to-use tool on the market, not only in DNA sequencing, but also in monitoring environmental pollution and ultra-sensitive biomarker detection. We believe that, in the future, the fields of biomedical and bioengineering, biosensing, nanofluidics, and integrated sensing devices will benefit significantly from the nanopore technique.

## Figures and Tables

**Figure 1 nanomaterials-12-02384-f001:**
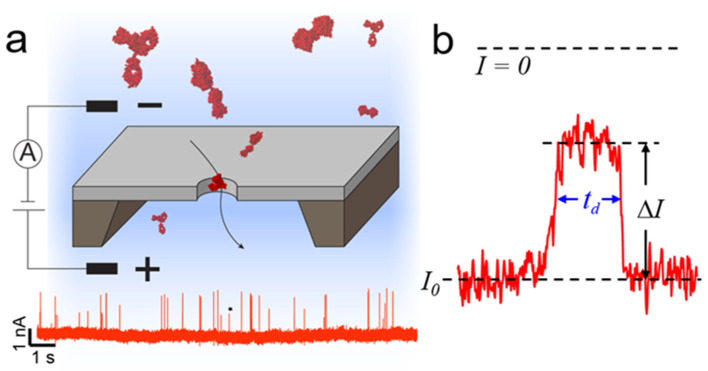
Single molecule detection by a synthetic nanopore. (**a**) Illustration of the concept of nanopore sensing. An electrical potential applied across a nanoscale pore creates a constant ionic current, and proteins passing through the sensing volume (indicated by the black arrow) produce resistive pulses. The red trace represents a typical measured current trace over time. Each spike represents a translocation of a protein through a nanopore. (**b**). Expanded trace (marked by * in panel (**a**)) representing a single protein translocating though a nanopore. The amplitude of current change, Δ*I*, is proportional to particle volume and the dwell time, *t_d_*, correlates to the particle charge and diffusion coefficient.

**Figure 2 nanomaterials-12-02384-f002:**
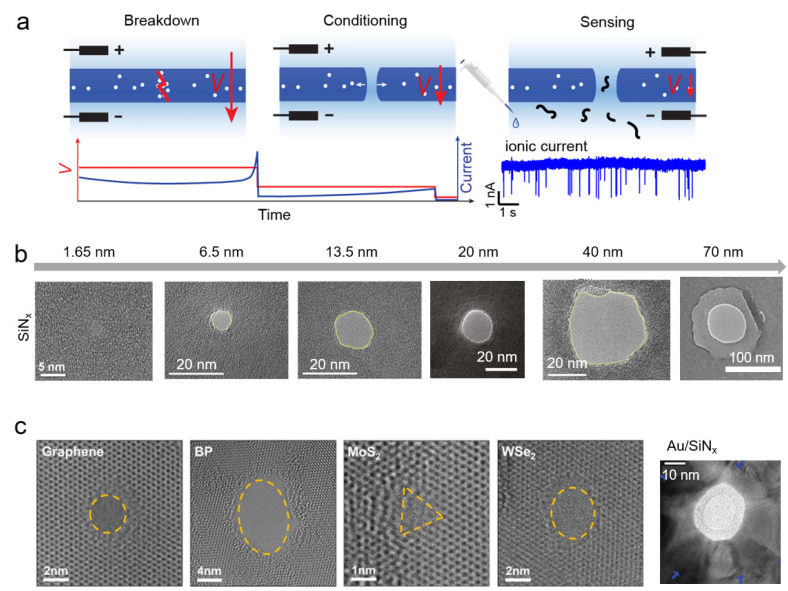
Controlled breakdown (CBD) applied in nanopore fabrication. (**a**) Schematic illustration of CBD-forming nanopore process Ref. [83] (not to scale). The red and blue curves represent the applied voltage and measured current, respectively. (**b**) CBD nanopores in SiN_x_ membrane ranging from 1.65 nm to 70 nm in diameter. Reprinted with permission from Ref. [10]. Copyright 2018 American Chemical Society. Reprinted with permission from Ref. [91]. Reprinted with permission from Ref. [118]. (**c**) CBD nanopores in different 2D membranes including Graphene, black phosphorus (BP), molybdenum disulfide (MoS_2_), and tungsten diselenide (WSe_2_), along with a nanopore formed in a gold-coated SiN_x_ membrane. Reprinted with permission from Ref. [119]. Copyright 2018 American Chemical Society. Adapted with permission from [120]. Copyright 2014 John Wiley & Sons, Inc.

**Figure 3 nanomaterials-12-02384-f003:**
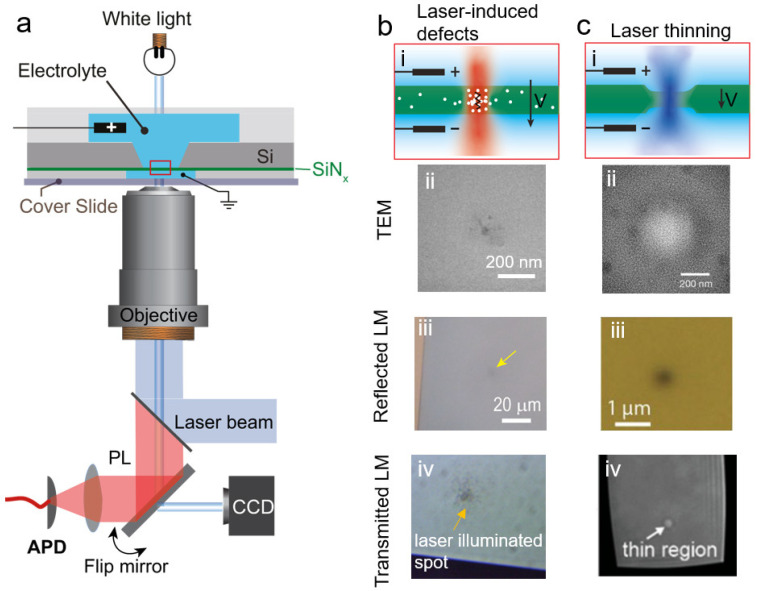
Laser-assisted breakdown to form nanopores locally. (**a**) Schematic of the experimental setup for laser-assisted breakdown. A collimated laser beam is focused on the membrane by an objective during the CBD process. The optical feedback signal, either PL intensity or Ca^2+^ flux fluorescence, is collected by the objective and received by a camera. Adapted with permission from Ref. [10]. Copyright 2018 American Chemical Society. (**b**) Laser-induced defect accumulation. The laser illumination enhances the defect density locally as (**i**) white dots. This defect accumulation appears as a dark spot in a TEM image (**ii**). This region has a lower reflection and transmission due to the high absorbance, therefore appears black in both reflected (**iii**) and transmitted (**iv**) light microscopy images. (**c**) Laser thinning. The focused laser illumination etches the membrane on both sides (**i**). The thinned region appears white in a TEM image (**ii**). Light microscopy images show a dark spot in reflection (**iii**) and a white spot in transmission (**iv**). Adapted with permission from Ref. [10]. Copyright 2018 American Chemical Society. Adapted with permission from Ref. [88]. Adapted with permission from Ref. [89]. Copyright 2018 American Chemical Society.

**Figure 4 nanomaterials-12-02384-f004:**
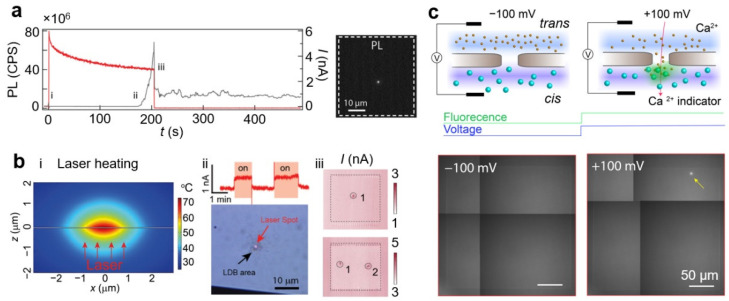
Additional approaches to provide feedback control in CBD. (**a**) Photoluminescence (PL) intensity provides the information of the membrane thickness. The PL intensity decreases over time indicating gradually reduced thickness of the membrane. A sudden drop of the PL intensity suggests the formation of the nanopore. Adapted with permission from Ref. [88]. (**b**) (**i**) Temperature profile of a 30 nm thick silicon nitride membrane when illuminated by a focused laser beam at 12 mW. (**ii**) The ionic current increases when the laser is focused on the nanopore due to the decreased viscosity at high temperatures. This increase in conductivity is reversible by turning the laser on and off. (**iii**) The ionic conductivity map when scanning laser position. The dashed circles highlight the regions where nanopores are located. Adapted with permission from Ref. [10]. Copyright 2018 American Chemical Society. Adapted with permission from Ref. [87]. Copyright 2021 American Chemical Society. (**c**) Schematic illustration showing that the Ca^2+^ gradient and Ca^2+^-indicator report nanopore position. The cis chamber contains Fluo-8 and the trans chamber contains Ca^2+^ ions. The fluorescence intensity increases locally when the Ca^2+^ ions are electrophoretically driven through a nanopore (+100 mV). The fluorescence microscopy images on the bottom show a bright spot at the nanopore position (indicated by a yellow arrow) at positive applied potential. Adapted with permission from Ref. [10]. Copyright 2018 American Chemical Society.

**Figure 5 nanomaterials-12-02384-f005:**
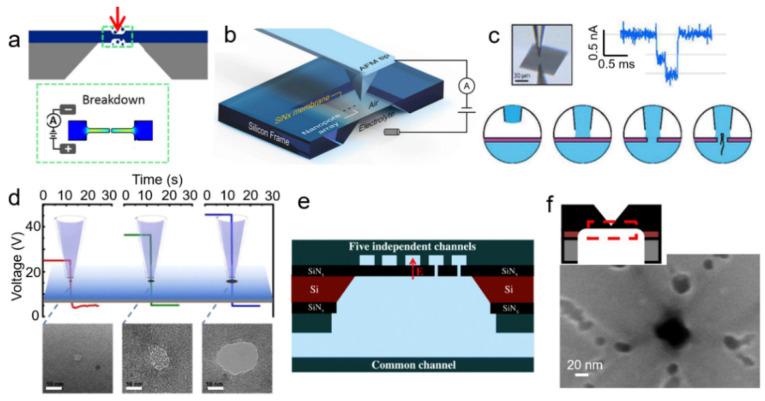
Tip-induced breakdown. (**a**) Schematic of a thinned area in a SiN_x_ membrane introduced via helium-ion beam. The inset in the green box shows the localized breakdown at the pre-thinned region. Adapted with permission from Ref. [150]. Copyright 2017 IOP Publishing. (**b**) 3D schematic of an experimental setup for AFM tip-controlled local breakdown (TCLB). Nanopore arrays are formed by controlling the position and the voltage of the AFM tip. Reprinted with permission from Ref. [153]. Copyright 2019 John Wiley & Sons, Inc. (**c**) Schematic of experimental setup of local breakdown introduced by micro-liquid conduction using a micropipette. Adapted with permission from Ref. [154]. Copyright 2017 American Chemical Society. (**d**) Transient high electric-field breakdown by the meniscus contact of a micropipette with the SiNx membrane [151,152]. Figures reprinted with permission from Ref. [151]. Scale bars are 10 nm. (**e**) A five-channel microfluidic system allows the formation of nanopores at precise locations on the membrane by controlling the electric field and electrolyte [155,156]. Reprinted with permission from Ref. [155]. (**f**) A schematic illustration of the tip of an inverted pyramid along with an SEM image showing a nanopore formed in the tip region. Adapted with permission from Ref. [196]. Copyright 2021 Elsevier.

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
