# Peer review of "Localized Nanopore Fabrication via Controlled Breakdown"

_nanomaterials, 2022, doi:10.3390/nano12142384_

Round 1
Reviewer 1 Report
This paper is a very good review on the topic of nanopore fabrication by means of controlled dielectric breakdown. The authors introduced clearly the topic, still highly interesting for the community, and reported all the major examples of methods based on dielectric breakdown to prepare nanopores to be used in single molecule detection.
I have only few minor comments:
- line 44: "thereby reavling"..."thereby revealing"
- section 2.2.1: material composition of the membrane. here the authors discussed only the case of SiNx while other materials are possible
- line 346-349. the sentence is a repetition of something already said above
Author Response
We thank the referee for the valuable input and positive assessments of our manuscript. We believe that the manuscript has benefitted from their referees’ suggestions and comments. Our responses are here in blue.

Reviewer 2 Report
I think the "Conclusion and Outlook" section could be expanded. The reviewed methods all solve certain problems with localized nanopore formation, but what are the remaining obstacles in general?
Author Response
We thank the referees for their time, valuable input and positive assessments of our manuscript. We believe that the manuscript has benefitted from their referees’ suggestions and comments. Our responses are here in blue in the attached file.

Reviewer 3 Report
The manuscript entitled “Localised Nanopore Fabrication via Controlled Breakdown” by Cuifeng Ying et al reviews the approaches employed for controlling the localization of synthetic nanopores produced by controlled breakdown. This may prove fruitful for accelerating their development as traditional single molecule sensors by resistive pulse techniques, and it is anticipated to substantially improve their applicability by facilitating optical measurements.
After the introductory section, the review is focused on the principles of nanopore fabrication by controlled breakdown (section 2), followed by a description of current approaches for improved localization. The proposed topic is of interest to a broad audience, the most important elements of a review in the field are satisfactorily covered, and the references are relevant. My concerns with respect to this review are minor, and they are presented next.
1. The term “recent” is a relative one; however, an eight-year-old development is not necessarily recent in the field of nanopores. I would recommend replacing “recently developed methods…” with “one of the latest developed methods – controlled breakdown…” at line 19 in the abstract.
2. Biological nanopores have been first proposed and used for single molecule characterization and sequencing; many references include information on their use, yet the review does not properly mention them. I would recommend a brief mentioning of biological nanopores and the advantages/disadvantages presented by synthetic nanopores. This will strengthen the scientific message of the review, which is focused on synthetic nanopores.
3. In the same context, Figure 1a shows the principle of detection with synthetic nanopores yet the main text (line 34-35) describes a generic “nanopore sensor”. Nanopore sensors include biological nanopores reconstituted in lipid membranes and their diagram would be quite different although the measuring principles are the same. The authors may indicate in the main text (line 34) that the principles of measurements are solely descriptive of a synthetic nanopore.
4. The terms “conducting ions” and “volume of conducting ions” (line 39 and line 44) may be confusing; the ions themselves do not conduct, they only contribute to conductivity. I would recommend using “blocks (or impedes) the ionic flow” at line 39, and “electrolyte solution volume displaced by the particle” at line 44 (or something similar).
5. The manuscript has numerous typos and formatting issues. For example, et al is not an abbreviation so no period is needed (unless it is at the end of a sentence), and "in situ" is not hyphenated when it is used as an adverb. Addition of references after the punctuation signs and without any space is very unusual formatting, and all the references from the same groups should be kept together (e. g., the end of the caption from Figure 3). I would recommend the authors to correct all the typos and follow the instructions for authors when formatting the manuscript.
I consider that the manuscript satisfies the criteria for publication as a review in Nanomaterials contingent to addressing the minor issues mentioned above.
Author Response

(The authors gave the same response as above.)

Reviewer 4 Report
This contribution of Ying and coauthors to Nanomaterials is an outstanding and very documented review of "Localised Nanopore Fabrication via Controlled Breakdown". They present in details the experimental technique, its advantages as well as its drawbacks and challenges.
More than 190 references are included in the bibliography, making this article a major starting point for scientists newly involved in the research.
Most of the work seems currently to be carried out on SiNx membranes but the authors quote other materials. I would simply suggest to had some deeper discussion on those materials, presenting their advantages and drawbacks in more details.
Author Response
We thank the referees for their time, valuable input and positive assessments of our manuscript. We believe that the manuscript has benefitted from their referees’ suggestions and comments. Our responses are in blue in the attached file.
